# Blocking Gi/o-Coupled Signaling Eradicates Cancer Stem Cells and Sensitizes Breast Tumors to HER2-Targeted Therapies to Inhibit Tumor Relapse

**DOI:** 10.3390/cancers14071719

**Published:** 2022-03-28

**Authors:** Cancan Lyu, Yuanchao Ye, Ronald J. Weigel, Songhai Chen

**Affiliations:** 1Department of Neuroscience and Pharmacology, Carver College of Medicine, University of Iowa, Iowa City, IA 52242, USA; cancan-lyu@uiowa.edu (C.L.); yuancye@uw.edu (Y.Y.); 2The Department of Surgery, Carver College of Medicine, University of Iowa, Iowa City, IA 52242, USA; ronald-weigel@uiowa.edu; 3The Holden Comprehensive Cancer Center, Carver College of Medicine, University of Iowa, Iowa City, IA 52242, USA

**Keywords:** cancer stem cells, G protein-coupled receptors, HER2+ breast cancer, HER2-targeted therapy, tumor relapse

## Abstract

**Simple Summary:**

Cancer stem cells (CSCs) are associated with therapeutic resistance and tumor relapse but effective approaches for eliminating CSCs are still lacking. The aim of this study was to assess the role of G protein-coupled receptors (GPCRs) in regulating CSCs in breast cancer. We showed that a subgroup of GPCRs that coupled to Gi/o proteins (Gi/o-GPCRs) was required for maintaining the tumor-forming capability of CSCs in HER2+ breast cancer. Targeting Gi/o-GPCRs or their downstream PI3K/AKT and Src pathways was able to enhance HER2-targeted elimination of CSCs and therapeutic efficacy. These findings suggest that targeting Gi/o-GPCR signaling is an effective strategy for eradicating CSCs, enhancing HER2+ targeted therapy and blocking tumor recurrence.

**Abstract:**

Cancer stem cells (CSCs) are a small subpopulation of cells within tumors that are resistant to anti-tumor therapies, making them a likely origin of tumor relapse after treatment. In many cancers including breast cancer, CSC function is regulated by G protein-coupled receptors (GPCRs), making GPCR signaling an attractive target for new therapies designed to eradicate CSCs. Yet, CSCs overexpress multiple GPCRs that are redundant in maintaining CSC function, so it is unclear how to target all the various GPCRs to prevent relapse. Here, in a model of HER2+ breast cancer (i.e., transgenic MMTV-Neu mice), we were able to block the tumorsphere- and tumor-forming capability of CSCs by targeting GPCRs coupled to Gi/o proteins (Gi/o-GPCRs). Similarly, in HER2+ breast cancer cells, blocking signaling downstream of Gi/o-GPCRs in the PI3K/AKT and Src pathways also enhanced HER2-targeted elimination of CSCs. In a proof-of-concept study, when CSCs were selectively ablated (via a suicide gene construct), loss of CSCs from HER2+ breast cancer cell populations mimicked the effect of targeting Gi/o-GPCR signaling, suppressing their capacity for tumor initiation and progression and enhancing HER2-targeted therapy. Thus, targeting Gi/o-GPCR signaling in HER2+ breast cancer is a promising approach for eradicating CSCs, enhancing HER2+ targeted therapy and blocking tumor reemergence.

## 1. Introduction

Cancer stem cells (CSCs), sometimes referred to as tumor initiating cells (TICs), are a small subpopulation of cells within tumors that possess characteristics of normal stem cells, i.e., the capacity of self-renewal and differentiation into the heterogeneous cell types [1,2]. Functionally, CSCs are defined by their ability to form tumorspheres in nonadherent conditions in vitro, to initiate a tumor from a few cells in vivo, and to self-renew to give rise to a secondary tumor when passaged. Compared to the bulk tumor cells, CSCs are relatively resistant to many therapies, including chemotherapy, radiotherapy, and immunotherapy, and thus are the leading cause of tumor recurrence and metastasis [1].

First discovered in acute myelogenous leukemia, CSCs are detectable in many solid tumor types, including breast cancer [3]. Cells expressing CSC markers (e.g., CD44^+^/CD24^−^) and with high aldehyde dehydrogenase (ALDH) activity were identified as a subset of cells in both breast cancer cell lines and human tumor tissues that could self-renew and initiate and maintain tumor growth [4,5]. Notably, the aggressive human HER2+ breast cancer subtype is enriched with CSCs, suggesting they contribute to the poor clinical prognosis for HER2+ breast cancer, one prone to recurrence and metastasis [6,7,8]. Indeed, HER2 overexpression can drive CSC expansion, and the effectiveness of HER2-targeted therapy (e.g., trastuzumab) is linked to the capacity to eradicate CSCs [6,9,10,11]. Moreover, trastuzumab resistance is attributed in part to expansion of the transtuzumab-resistant CSC population [11,12]. These findings suggest CSCs represent an attractive target for developing new HER2+ breast cancer therapies that overcome drug resistance and achieve more durable remission.

Diverse signaling pathways regulate CSC self-renewal and tumorigenicity. In HER2+ breast CSCs, hyperactivation of HER2 can directly activate signaling pathways, including PI3K/AKT and Src [6,13,14], and HER2 can interact with several other dysregulated pathways including those downstream of Wnt, Notch, hedgehog, integrin, NFkB, and G protein coupled receptors (GPCRs) [15,16,17]. GPCRs are the largest family of cell surface receptors (over 800 members) and transmit signals through diverse G proteins, including G_i/o_, G_s_, G_q/11_, and G_12/13_ [18,19]. GPCRs play a central role in regulating the function of CSCs [16,17]. In HER2+ breast cancer, CSC expansion and tumor growth and metastasis is stimulated by several chemokines, including CCL2 and interleukin 8, which stimulate overexpressed cognate receptors [20,21]. Many GPCRs appear to be overexpressed by HER2+ breast CSCs; however, except for a few GPCRs, the role of GPCR signaling in CSC function and response to therapy is unclear [22]. Moreover, how to target the multiplicity of signaling GPCRs and overcome their potential redundant functions remains unexplored.

Gi/o-coupled GPCRs (Gi/o-GPCRs) are the largest subgroup of nonsensory GPCRs that have known G protein coupling [23]. Activation of these receptors primarily inhibit adenylyl cyclase via Gαi/o proteins, decreasing cAMP production, which, in turn, decreases the activity and function of cAMP-dependent protein kinase [24]. Additionally, Gi/o-GPCRs may regulate diverse cellular processes including cell proliferation and transformation through other signaling pathways such as Src and PI3K via Gαi/o proteins or Gβγ proteins liberated from the activated Gαi/o proteins [24,25,26]. Notably, many Gi/o-GPCRs, such as PAR1, LPA and chemokine receptors, play an important role in regulating cancer development and progression [25,27,28,29,30]. We recently found that HER2 overexpression and hyperactivation in breast cancer cells alters GPCR expression, leading to aberrant signaling by Gi/o-GPCRs [23]. In a mouse model of HER2+ breast cancer, i.e., MMTV-Neu (Neu), which express an activated rat ErbB2/HER2 homologue selectively in the mammary gland [31], tumor onset, growth, and metastasis were suppressed by blocking Gi/o-GPCR signaling, either via a catalytic subunit (the A-protomer) of pertussis toxin (which selectively uncouples Gi/o proteins from their cognate receptors) or inhibitors that target the PI3K/AKT and Src downstream effectors [23]. A Gi/o-GPCR signaling blockade also enhanced HER2-targeted therapy in this model system [23]. These data imply that targeting Gi/o-GPCR signaling might be an effective way to therapeutically eradicate CSCs.

In this study, we tested the role of Gi/o-GPCR signaling in CSC function in HER2-induced breast cancer, using genetic mouse models and breast cancer cell lines. Moreover, we determined the effect of eliminating CSCs on HER2-induced tumor development and response to HER2-targeted therapy. Our results demonstrate that Gi/o-GPCRs signal through the PI3K/AKT and Src pathways to promote the tumorigenicity of CSCs and modulate their sensitivity to HER2-targeted therapy. These findings suggest that targeting the downstream signaling pathways shared by multiple Gi/o-GPCRs is an effective strategy for eradicating CSCs, and in turn blocking tumor development and enhancing HER2-targeted therapy.

## 2. Materials and Methods

### 2.1. Reagents

PTx was from Sigma (St. Louis, MO, USA). Trastuzumab was from Genetech (South San Francisco, CA, USA). Laptinib, GDC0941 and saracatinib were from LC Laboratories (Woburn, MA, USA). Antibodies for AKT (no. 4685), phospho-AKT^S473^ (no. 4060), Src (no. 2109), and phospho-Src^Y416^ (no. 6943) were from Cell Signaling Technology (Danver, MA, USA); GADPH (sc-47724) was from Santa Cruz Biotechnology (Dallas, TX, USA). Antibodies for mouse FITC-conjugated CD24 (#101806), allophycocyanin-conjugated CD49f (#313616), and phycoerythrin-conjugated CD61 (#104308) were from Biolegend (San Diego, CA, USA). GCV was from TCI America (Portland, OR, USA).

### 2.2. Mouse Studies

All animal studies were conducted in accordance with an Institutional Animal Care and Use Committee-approved protocol at the University of Iowa. The transgenic mice used in this study were described previously [23]. All mice were in the FVB/N genetic background. Female mice were kept as virgins throughout the experiments.

To test the tumorigenicity of CSCs in premalignant mammary glands, LP and basal cells were isolated from the mammary glands of 4.5-month-old Neu and Neu/PTx mice via fluorescence-activated cell sorting (FACS). A total of 10,000 LP and 5000 basal cells were injected into the inguinal mammary fat pads of nude mice. Tumor formation was monitored weekly via palpation, and the tumor size was measured using a caliper.

To determine the ability of primary tumors to give rise to secondary tumors, tumor cells were isolated from size-comparable tumors arisen from Neu and Neu/PTx transgenic mice, using a mammary epithelial cell enrichment kit (StemCell Technologies; Vancouver, Canada). Then, 1 × 10^5^ cells were injected into the right inguinal mammary fat pads of nude mice. Tumor growth was monitored weekly via palpation and caliper measurement.

To determine the effect of eliminating CSCs on tumor initiation and progression, 1 × 10^6^ Neu cells expressing SORE6-mCherry or SORE6-hTK were injected into the inguinal mammary fat pads of nude mice and then treated with GCV (25 mg/kg, i.p., once daily) for 21 days immediately following Neu cell implantation. To determine how tumor progression is affected by lapatinib or lapatinib in combination with GCV, the mice were treated with lapatinib (200 mg/kg, o.g, once daily) or lapatinib plus GCV (25 mg/kg, i.p., once daily) for 3 weeks when the tumors reached a size of ~150–200 mm^3^. The tumor growth was recorded with a caliper measurement of the tumor length (L) and width (W), every 5 days. The tumor volume was calculated using the formula of length × width^2^ × 0.5.

### 2.3. Fluorescence-Activated Cell Sorting

The premalignant mammary glands from 4.5-month-old Neu and Neu/PTx mice were dissociated using a mammary epithelial cell enrichment kit (StemCell Technologies). The dissociated single cells were first stained with biotin-labeled anti-mouse CD31, CD45, and Ter119, followed by PE/Cy7-cojugated streptavidin, FITC-conjugated anti-mouse CD24, APC-conjugated anti-mouse CD49f, and PE-conjugated anti-mouse CD61. The luminal (CD24^high^CD49f^low^CD61^−^), luminal progenitor (CD24^high^CD49f^low^CD61^+^), and basal cell (CD24^med^CD49f^high^) subpopulations were sorted from the Lin^-^ epithelial cell population (CD45^−^CD31^−^Ter119^−^) using an Aria II cell sorter (BD Bioscience; San Jose, CA, USA) with over 95% purity.

### 2.4. Flow Cytometry

Subconfluent cultured cells were dissociated with Accutase and resuspended in PBS for flow cytometry analysis of CD49f and CD61 expression or SORE6-mCherry expression as we described previously [32]. To determine the effect of drug treatment, cells were treated with drugs for four days followed by one-day culture in the absence of drugs prior to flow cytometry analysis.

### 2.5. Cell Lines

Neu cells were generated from tumors arisen from the transgenic mice, MMTV-c-Neu, and cultured in DMEM media containing 10% FCS. BT474 cells were purchased from the ATCC. The trastuzumab-resistant BT474 derivative (BT474R) cells were kindly provided by Dr. Hank Qi (Department of Anatomy and Cell Biology, University of Iowa, Iowa City, Iowa). Cell lines were tested for *Mycoplasma* using a mycoplasma detection kit (ATCC). BT474 and BT474R cells were cultured in DMEM/F12 media containing 10% FCS. Each cell line was cryopreserved at low passage numbers (less than six passages after receipt) and used in experiments for a maximum of 18 passages.

### 2.6. Isolation of Mammary Epithelial Cells

Mammary epithelial cells were isolated from the mammary glands of 4.5-month-old Neu and Neu/PTx transgenic FVB/N mice, using an EasyStep Mouse Epithelial Cell Enrichment Kit (StemCell Technologies).

### 2.7. Lentiviral Constructs

Lentiviral vectors for expressing the SORE6-driven destabilized copepod mCherry (SORE6-mCherry) and its control, the minimal CMV promotor-driven mCherry (mCMV-mCherry), tetracycline-inducible expression of myristoylated constitutively active AKT2 mutant, GFP-tagged, constitutively active Src mutant, and Src^Y527F^ were described in previous studies [25,32]. Lentiviruses were generated in HEK293FT cells and concentrated using a Lenti-X Concentrator (Takara Bio; San Jose, CA, USA).

### 2.8. Establishment of Stable Cell Lines

Neu, BT474, and BT474R cells were transduced with lentiviruses encoding mCMV-mCherry or SORE6-mCherry and selected with hygromycin (1 mg/mL) for at least 1 week to establish stable lines. Neu cells stably expressing tetracycline-inducible myristoylated AKT2 and Src^Y527F^ were described previously [23].

### 2.9. Tumorsphere Culture

Single cells were plated on ultralow attachment, 12- or 6-well plates for tumorsphere formation, as described previously [32]. After 7 to 14 days of culture, the tumorspheres were counted under an inverted microscope. To measure the size of the tumorspheres, images were taken at 5–10 randomly chosen areas under a phase contrast microscope and analyzed using ImageJ software.

### 2.10. Western Blotting Analysis

Protein lysates were prepared from cells and tumor tissues and analyzed via Western blotting as we described, using the iBright CL1000 (Thermo Fisher Scientific; Waltham, MA, USA) or Odyssey (LI-COR Biotechnology; Lincoln, NE, USA) imaging system [25,33]. The uncropped blots and molecular weight markers are shown in Appendix A.

### 2.11. Statistics

Data were expressed as mean ± SEM. Statistical comparisons between groups were analyzed using a two-tailed Student’s *t* test or ANOVA (*p* < 0.05 was considered significant). The survival curves were analyzed according to the Kaplan–Meier method.

## 3. Results

### 3.1. Blocking Gi/o-GPCR Signaling Suppresses the Tumorigenicity of HER2-Induced CSCs

To test whether Gi/o-GPCR signaling regulates HER2-induced CSCs, Gi/o signaling was specifically targeted with the catalytic subunit (the A-protomer) of PTx [34], in a model of breast cancer. Here, pre-malignant mammary glands were isolated from 4.5-month-old transgenic mice with mammary glands expressing Neu and a catalytic subunit of PTx under doxycycline control (Neu and MMTV-tTA/TetO-PTx/MMTV-Neu; Neu/PTx). Mammary glands were dissociated, and the luminal, luminal progenitor (LP), and basal cell subpopulations were sorted from the Lin^-^ epithelial cell population (CD45^−^CD31^−^Ter119^−^), using the established cell surface markers CD24^high^CD49f^low^CD61^−^, CD24^high^CD49f^low^CD61^+^, and CD24^med^CD49f^high^ (Figure 1A) [35,36,37,38]. PTx expression did not significantly change the number of luminal, luminal progenitor, or basal cells (Figure 1B) but significantly suppressed tumorspheres from forming from LP and basal cells in ultra-low adhesive plates, by ~60% and ~85%, respectively (Figure 1C), and at a significantly reduced size (Figure 1D). These findings suggest Gi/o-GPCR signaling maintains stemness in HER2-induced CSCs. To validate these results in vivo, freshly isolated LP and basal cells were transplanted into the mammary glands of nude mice and tumor initiation and progression was monitored. As shown in Figure 2A,B, although tumor onset from LP cells was similar between Neu and Neu/PTx mice, tumors generated from LP cells of Neu/PTx mice grew more slowly. In contrast, basal cells from Neu/PTx mice were significantly delayed in tumor onset and proliferated more slowly, compared to those from Neu mice (Figure 2C,D).

To test if Gi/o-GPCR signaling contributes to CSC maintenance in established tumors, we compared cells isolated from Neu vs. Neu/PTx mice for the capacity to form tumorspheres. As shown in Figure 3A,B, cells isolated from established tumors of Neu/PTx mice formed significantly fewer tumorspheres, suggesting Neu/PTx tumors contained fewer CSCs. Moreover, when the same number of tumor cells (1 × 10^5^) was implanted into the mammary gland of nude mice, tumor onset was significantly delayed and tumors grew slower in mice injected with Neu/PTx tumor cells (Figure 3C,D). These results indicate that blocking Gi/o-GPCR signaling is an effective strategy for suppressing the self-renewal activity of CSCs in primary tumors that give rise to secondary tumors.

### 3.2. Blocking Gi/o-GPCR Signaling Enhances CSC Sensitivity to HER2-Targeted Therapy

The effectiveness of HER2-targeted therapy is linked to its ability to eradicate breast CSCs [6,9,10,11]. To test if Gi/o-GPCR signaling drives CSCs resistance to HER2-targeted therapy, the effect of PTx treatment was tested in CSCs from two model systems: Neu cells, a primary tumor cell line from tumors of Neu mice, and human HER2+ breast cancer cell lines, BT474 and BT474R (a trastuzumab-resistant derivative of BT474 cells). PTx treatment reduced CSC populations in Neu cells that are marked by several established CSC markers (including CD49f^+^/CD61^+^ and the fluorescence reporter SORE6-mcherry that detects SOX2/OCT4-overexpressing CSCs) (Figure 4A,B) [39,40]. Similarly, in BT474 and BT474R cells, those CSC populations marked by SORE6-mCherry were also reduced by PTx (Figure 3C,D). Notably, lapatinib and trastuzumab treatment suppressed the SORE6-mCherry^+^ CSC population in Neu and BT474 cells (Figure 4B,C), but trastuzumab had little effect on the CSCs in BT474R cells (Figure 4D). Nevertheless, co-treatment with PTx enhanced the activity of lapatinib and trastuzumab to eliminate the CSCs in Neu and BT474 cells and restored the sensitivity of the CSCs in BT474R cells to trastuzumab (Figure 4B–D). Consistent with its activities in eradicating CSC populations, PTx treatment alone suppressed tumorsphere formation of Neu, BT474, and BT474R cells, and PTx enhanced lapatinib or transtuzumab suppression of tumorsphere formation in all cells tested (Figure 4E–H). These findings suggest Gi/o-GPCR signaling regulates CSC sensitivity to HER2-targeted therapy.

### 3.3. PI3K/AKT and Src Pathways Regulate CSC Sensitivity to HER2-Targeted Therapy

PI3K/AKT and Src signaling drives HER2-induced breast cancer progression and resistance to HER2-targeted therapy [41,42,43,44,45]. Previously, we showed these pathways signals are downstream arms of Gi/o-GPCR signaling that stimulate HER2+ breast cancer cell growth and migration, offering resistance to HER2-targeted reagents, including trastuzumab and lapatinib [23]. To test whether activation of PI3K/AKT and Src pathways confers CSC resistance, we blocked them with PI3K- and Src-specific inhibitors, GDC0942, and saracatinib at a concentration that selectively inhibits their activity [23]. Consistent with our PTx experiments, Neu and BT474R cells treated with GDC0942 and saracatinib contained far fewer SORE6-mcherry^+^ CSCs and generated fewer tumorspheres (Figure 5A–D). Combining the treatments did not enhance the effect, suggesting the PI3K/AKT and Src pathways are redundant in regulating CSC activity in HER2+ breast cancer cells (Figure 5A–D). GDC0942 and saracatinib also enhanced lapatinib efficacy in Neu cells and restored BT474R cell sensitivity to trastuzumab treatment, eradicating the CSC population and suppressing the tumorsphere-generating capacity (Figure 5A–D). Together, these findings suggest PI3K/AKT and Src signaling maintains CSC activity and constitutes a therapeutic target in HER2+ breast cancer cells.

To determine whether CSC tumorigenic activity relies on Gi/o-GPCR signaling through the downstream PI3K/AKT and Src pathways, we overexpressed a constitutively active mutant of either AKT2 (myristoylated AKT2) or Src (GFP-tagged Src^Y527F^) in Neu cells (Figure 5E). In Neu cells, overexpressing these constitutively active mutants boosted the SORE6-mcherry^+^ CSC populations (i.e., SORE6-mcherry^+^) and tumorsphere-generating capacity; it also largely abolished PTx toxicity to CSCs and tumorsphere formation (Figure 5F,G). Thus, CSC survival and self-renewal are likely supported by redundant Gi/o-GPCR signals to the AKT and Src pathways.

### 3.4. CSCs Elimination Suppresses Initiation and Progression by HER2+ Tumors and Enhances Lapatinib Therapeutic Efficacy

Manipulation of Gi/o-GPCR signaling in breast cancer cells likely affects both CSCs and non-CSCs. Thus, to show directly that CSCs drive the resistance to HER2+-targeted therapy, Neu cells were specifically targeted for CSC elimination by engineering them to express a suicide gene, in this case the human herpes simplex virus thymidine kinase (hTK), wherein hTK is expressed via the SORE6 promoter (i.e., SORE6-hTK) [32]. As expected, Neu cells expressing SORE6-hTK were sensitive to ganciclovir (GCV)-mediated killing and formed fewer tumorspheres than those expressing SORE6-mCherry (Figure 6A) [32,46]. To test if eliminating CSCs specifically impedes tumor formation, nude mice were implanted with Neu cells expressing SORE6-mcherry or SORE6-hTK. After 21 days, GCV (25 mg/kg, i.p.) potently inhibited tumor onset in injected cells that expressed SORE6-hTK (t_1/2_ was 30 days vs. 10 days; Figure 6B). Of mice injected with SORE6-mcherry-expressing cells (i.e., not expressing the suicide gene), 100% acquired tumors by 14 days after implantation. In contrast, even 38 days later, only 70% of mice injected with SORE6-hTK-expressing cells had developed tumors (Figure 6B), and tumors were significantly smaller (Figure 6C). These results corroborate our other data (Figure 2) showing CSCs contribute to Neu-induced tumor initiation and progression.

To determine whether eradicating CSCs could sensitize HER2+ breast cancer to HER2-targeted therapy, Neu cells expressing SORE6-mcherry or SORE6-hTK were implanted into nude mice. When tumors grew to a comparable size (~150 mm^3^), mice were treated with GCV (25 mg/kg) alone or lapatinib (100 mg/kg) plus GCV (25 mg/kg). GCV slowed tumor growth in mice injected with Neu cells expressing SORE6-hTK (Figure 6D). Combined treatment of GCV with lapatinib only partially suppressed tumor growth in mice injected with SORE6-mcherry-expressing Neu cells, but largely abolished and even caused regression of tumor growth in mice injected with SORE6-hTK-expressing cells (Figure 6D). These results indicate CSCs in tumors confer resistance to HER2-targeted therapy.

## 4. Discussion

Increasing evidence demonstrates that the aggressiveness, therapy-resistance, and disease-relapse exhibited by HER2+ breast cancer might be attributed to the presence of a small sub-population of cancer stem cells (CSCs) [6,7,8]. This study identifies Gi/o-GPCR signaling as a key activity driving CSC function in HER2-induced breast cancer, where HER2 overexpression by CSCs stimulates their tumorigenicity. In Neu transgenic mice, Neu-induced mammary cancer was initiated by two subpopulations of stem-cell-like mammary epithelial cells, identified as such by surface markers of basal cells or luminal progenitor cells (i.e., CD24^med^CD49f^high^ or CD24^high^CD49f^low^CD61^+^, respectively) [38]. For both populations, expression of the catalytic subunit (the A-protomer) of PTx impaired generation of tumorspheres in vitro and tumors in vivo. PTx is an A-B toxin consisting of A and B protomers [34]. Although PTx may exert Gi/o protein-independent action through the interaction of its B-oligomer with cell surface proteins, the A-protomer selectively ADP-ribosylates the α subunits of Gi/o proteins, resulting in the uncoupling of Gi/o proteins from their cognate receptors [47]. Thus, by using the PTx A-protomer to selectively block Gi/o-GPCR signaling, our findings suggest Gi/o-GPCRs are critical for CSC activity. Gi/o-GPCR signaling blockade also suppressed the ability of tumor cells from established primary Neu tumors to form tumorspheres in vitro and give rise to secondary tumors in vivo, suggesting Gi/o-CPCR signaling is also required for CSC-driven tumor progression. These results are consistent with our previous findings that tumor onset can be delayed and Neu-induced tumor growth and metastasis can be suppressed by blocking Gi/o-GPCR signaling with PTx [23]. Interestingly, in premalignant mammary glands of Neu mice, Gi/o-GPCR blockade had no effect on CSC populations. However, in vitro in several HER2+ breast cancer cell lines (i.e., Neu, BT474, and BT474R), Gi/o-CPCR blockade significantly reduced the CSC populations. These findings suggest that, although Gi/o-CPCR signaling supports CSC activity, other factors also contribute to breast CSC maintenance in vivo.

The presence of CSCs is thought to make HER2+ breast cancer resistant to HER2-targeted therapy, allowing tumors to re-emerge after treatment [8]. Consistent with this idea, we found trastuzumab and lapatinib only partially inhibited CSC activity in the therapy-sensitive cell lines, BT474 and Neu, while trastuzumab had no effect on CSCs in the trastuzumab-resistant cell line, BT474R. Moreover, ablating CSCs in Neu cells using the suicide gene construct, SORE6-hTK, not only suppressed tumor formation and growth but also enhanced the efficacy of lapatinib. Notably, blocking Gi/o-GPCR signaling in Neu and BT474 cells using PTx not only sensitized CSCs to lapatinib and trastuzumab but also restored the sensitivity of CSCs in BT474R cells to trastuzumab. These findings indicate that Gi/o-GPC signaling is crucial for conferring CSCs resistance to HER2-targeted therapy. Previously, we showed blocking Gi/o-GPCR signaling also enhances the sensitivity of the bulk of HER2+ breast cancer cells to HER2-targeted therapy, suggesting targeting Gi/o-GPCRs may have dual inhibitory effects on both CSCs and non-CSCs [23]. Since CSCs can drive therapy resistance and tumor relapse, co-targeting Gi/o-GPCRs might be able to enhance the efficacy of HER2-targeted therapy and prevent resistance.

Our results show that CSC maintenance in HER2+ breast cancer cells relies, at least in part, on Gi/o-GPCR signaling through the PI3K/AKT and Src pathways. Inhibiting either pathway with specific inhibitors at a concentration selectively blocking the relevant pathway mimicked Gi/o-GPCR blockade by PTx in suppressing CSC activity and sensitizing CSCs to HER2-targeted therapy in HER2+ breast cancer cells. Moreover, overexpressing an active mutant of either AKT2 or Src blocked the inhibitory effect of PTx on CSC activity, indicating the PI3K/AKT and Src pathways function, redundantly, downstream of Gi/o-GPCRs to regulate CSCs.

Previous studies showed aberrant PI3K/AKT and Src activation induces CSC expansion and trastuzumab resistance in HER2+ breast cancer [6,13,14]. These pathways appear to redundantly stimulate CSC sensitivity to HER2-targeted therapy, since tumorsphere reduction assays show inhibiting both pathways sensitizes them to trastuzumab and lapatinib better than inhibiting either alone. Notably, PI3K/AKT and Src are the key pathways activated downstream of both Gi/o-GPCRs and HER2 and the convergence points to crosstalk between Gi/o-GPCRs and EGFR/HER2 [44,45,48,49,50]. Targeting PI3K or AKT was shown to ablate CSCs and restore sensitivity of HER2+ breast cancer cells to trastuzumab [6,13]. Furthermore, dysregulated Src signaling was shown to contribute to resistance of HER2+ breast cancer to HER2-targeted therapy by regulating CSC stemness [14]. Our data demonstrate that inhibiting both PI3K/AKT and Src pathways is more effective in eradicating CSCs than inhibiting either pathway alone. These results may underlie our previous findings that combined inhibition of both PI3K and Src synergize to block tumor growth and sensitize tumors to HER2-targeted therapy [23].

The origin of CSCs remains largely unknown [1]. Recent studies suggest CSCs exhibit a high level of plasticity and have the ability to switch between CSC and non-CSC states [1,51]. Moreover, CSCs may consist of different sub-populations that can interconvert [1,51]. Our findings that Gi/o-GPCR blockade reduces CSC populations marked by different markers suggest Gi/o-GPCR signaling supports the maintenance of multiple CSC subsets. It remains to determine whether Gi/o-GPCR signaling regulates the dynamic switch between CSC and non-CSC states or different subsets of CSCs. Nevertheless, our functional assays demonstrate the importance of Gi/o-GPCR signaling in maintaining the stemness of CSCs for tumorigenesis and therapeutic resistance in HER2+ breast cancer. These findings have important practical implications for how to target GPCRs as a new therapeutic approach for eradicating CSCs in cancer and backs up considerable evidence that various GPCRs regulate CSCs. GPCR expression profiling shows that when somatic cells reprogram to CSCs, then 195 GPCRs are either up- or down-regulated [16]. In ALDH+ and ALDH− BT474 cell populations, comparing GPCR expression also found 19 GPCRs were overexpressed in the CSC population [22]. Activation of several Gi/o-coupled chemokine receptors, including CCR2, CXCR1, and CXCR2, by their cognate ligands was found to stimulate CSC activity in HER2+ breast cancer [20,21]. Although blocking the interaction of interleukin 8 with CXCR1 and CXCR2 enhanced lapatinib-inhibited CSC activity and suppressed tumor growth in a xenograft mouse model, concern remains that targeting individual GPCRs may not be effective for eradicating CSCs [21,52]. Since multiple GPCRs regulate CSC activity, targeting any one may lead to compensation by another. Indeed, attempts to target individual GPCRs as a cancer therapy have failed in many clinical trials, largely due to a lack of efficacy [53]. Although GPCRs serve as a target for nearly 40% of the drugs on the current market, only a few were effective for cancer [53]. Our findings provide a potential solution to overcoming this problem: target the downstream pathways shared by multiple Gi/o-GPCRs, such as the PI3K/AKT and Src pathways, which are critical for CSC maintenance and self-renewal.

## 5. Conclusions

In summary, our studies provide the first evidence that a subgroup of GPCRs, Gi/o-GPCRs, is critical for CSC function in HER2+ breast cancer and that targeting signaling by Gi/o-GPCRs through PI3K/AKT and Src pathways enhances HER2-targeted therapy for eradicating CSCs. Given that dysregulation of multiple GPCRs in CSCs is a common phenomenon not only in HER2+ breast cancer but also other cancers and breast cancer subtypes [16,17], our findings may have important implications for novel approaches that target GPCRs as a new way to eradicate CSCs. Since CSCs drive drug resistance and tumor relapse, translation of these findings into clinical trials should identify a more effective strategy that can improve the efficacy of current anticancer therapies.

## Figures and Tables

**Figure 1 cancers-14-01719-f001:**
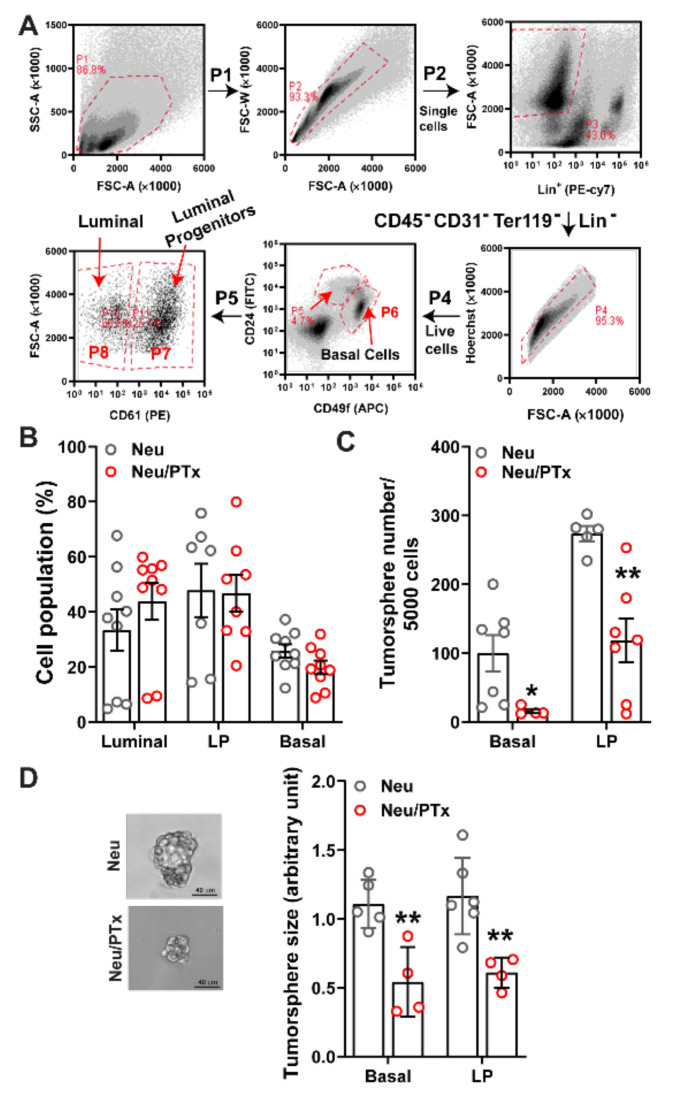
Blocking Gi/o-GPCR signaling in Neu-induced breast CSCs inhibits tumorsphere outgrowth. (**A**) Schematic representation of isolating (via flow cytometry) luminal, luminal progenitors (LPs), and basal cells (basal) from premalignant mammary glands of mice. (**B**–**D**) The effect of PTx expression on luminal, LP, and basal populations (**B**) in Neu transgenic mice, and the size (**C**) and number (**D**) of tumorspheres formed from basal and LP cells in vitro. Representative images of tumorspheres from the basal cells of Neu and Nue/PTx mice are shown on the left panel of (**D**). *, ** *p* < 0.05 and 0.01 vs. Neu, respectively, *n* = 4–7.

**Figure 2 cancers-14-01719-f002:**
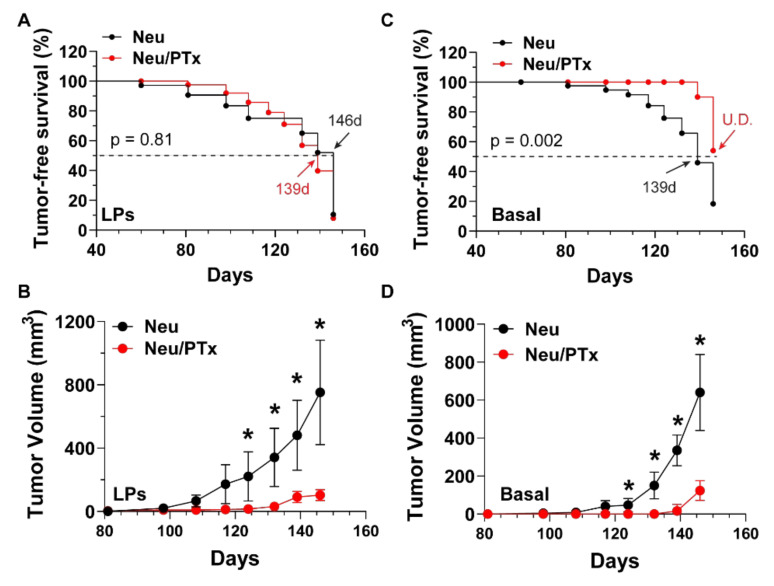
Blocking Gi/o-GPCR signaling suppresses the tumor-forming capacity of breast CSCs in vivo. A total of 10,000 and 5000 of luminal progenitor (LPs; (**A**,**B**)) and basal (basal; (**C**,**D**)) cells isolated from Neu and Neu/PTx mice were orthotopically implanted into the mammary gland of nude mice, and tumor growth was measured via caliper. (**A**,**C**) Tumor-free survival curves; (**B**,**D**) tumor growth curves. * *p* < 0.05 vs. Neu, *n* = 5.

**Figure 3 cancers-14-01719-f003:**
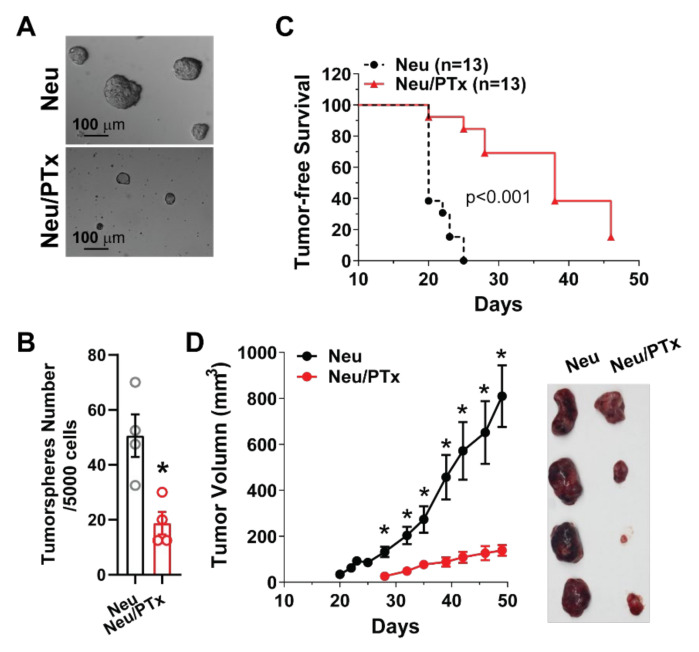
Gi/o-GPCR signaling regulates CSC tumorigenecity in established breast tumors. Total 5000 cells were plated per well. Cells were plated on 6-well plates for tumorsphere-forming assays. Representative images of tumorspheres are shown in (**A**), and quantitative data of tumorsphere number are shown in (**B**). * *p* < 0.05 vs. Neu, *n* = 4. (**C**,**D**) 1 × 10^5^ cells were implanted into the mammary gland of nude mice, and tumor growth was measured via caliper. Tumor-free survival curves are shown in (**C**), and tumor growth curves are shown in (**D**). Representative images of tumors are shown in the right panel of (**D**). * *p* < 0.05 vs. Neu/PTx, *n* = 8–10.

**Figure 4 cancers-14-01719-f004:**
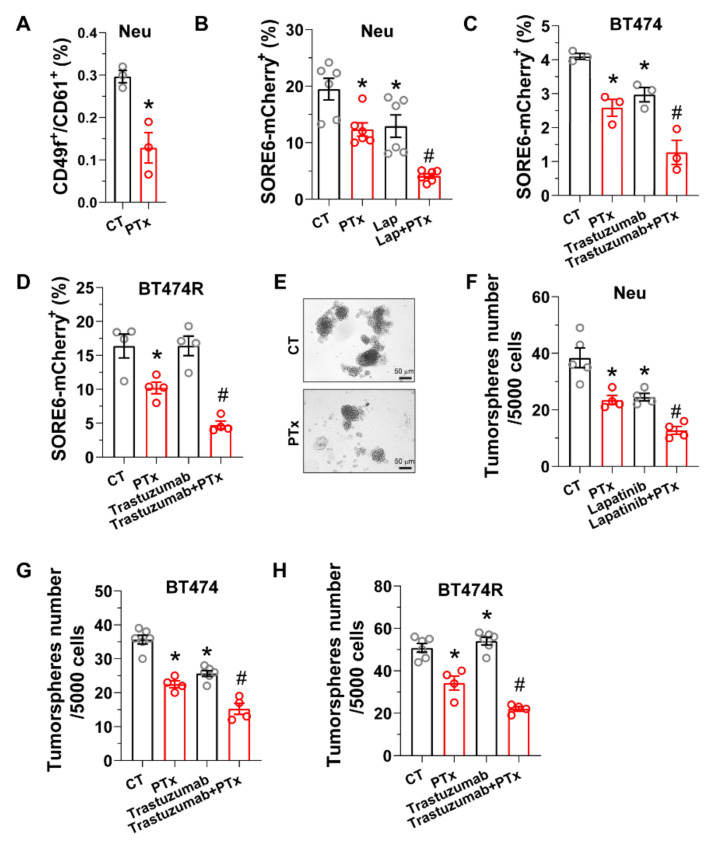
Gi/o GPCR signaling regulates the sensitivity of breast CSCs to HER2-targeted therapy. Neu, BT474, and BT474R cells were treated with PTx (0.2 μg/mL), trastuzumab (1 μg/mL), and lapatinib (1 μM), either alone or in combination. The effect on (**A**–**D**) CSC populations was determined via flow cytometry; (**E**–**H**) tumorsphere formation was determined via tumorsphere-formation assays. (**E**) Representative images of tumorspheres for Neu cells. * *p* < 0.05 vs. CT, *n* = 3–6. # *p* < 0.05 vs. PTx, lapatinib, or trastuzumab alone, *n* = 4–6.

**Figure 5 cancers-14-01719-f005:**
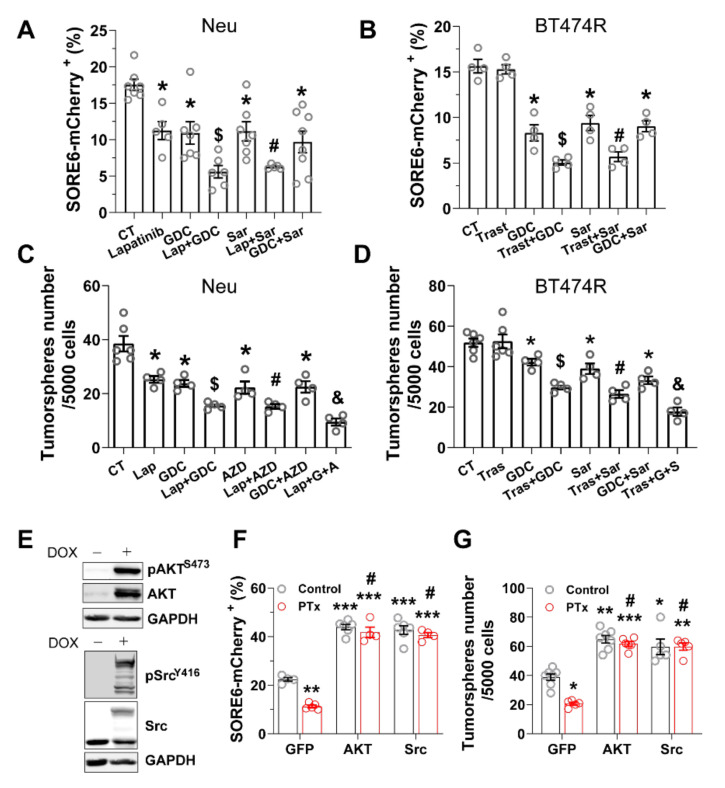
Gi/o-GPCRs signal through the PI3K/AKT and Src pathways to maintain CSC activities in HER2+ breast cancer cells. (**A**–**D**) Inhibiting PI3K and Src activities in Neu and BT474R cells reduces CSC populations (**A**,**B**) and tumorsphere formation (**C**,**D**). Cells were treated with lapatinib (1 μM), GDC0941 (GDC, 0.1 μM), saracatinib (Sar, 0.1 μM), and trastuzumab (1 mg/mL), either alone or in combination, for 3 days for CSC analysis and 6 days for tumorsphere-formation assays. * *p* < 0.05 vs. CT; $ *p* < 0.05 vs. lapatinib, trastuzumab, and GDC alone; # *p* < 0.05 vs. lapatinib, trastuzumab, and Sar alone; *p* < 0.05 vs. lap + GDC, lap + Sar, Tras + GDC, Tras + Sar, and GDC + Sar. *N* = 4–7. (**E**) Western blotting showing induced expression of myristoylated AKT2 and GFP-tagged Src^Y527F^ mutants in Neu cells, after 3 days of doxycycline (2 μg/mL) treatment. (**F**–**G**) The effect of overexpressing active AKT2 and Src mutants on the response of Neu cells to PTx-mediated inhibition of CSC populations (**F**) and tumorsphere formation (**G**). *, **, *** *p* < 0.05, 0.01 and 0.001 vs. GFP control, respectively; # not significantly different vs. control. *N* = 4–6.

**Figure 6 cancers-14-01719-f006:**
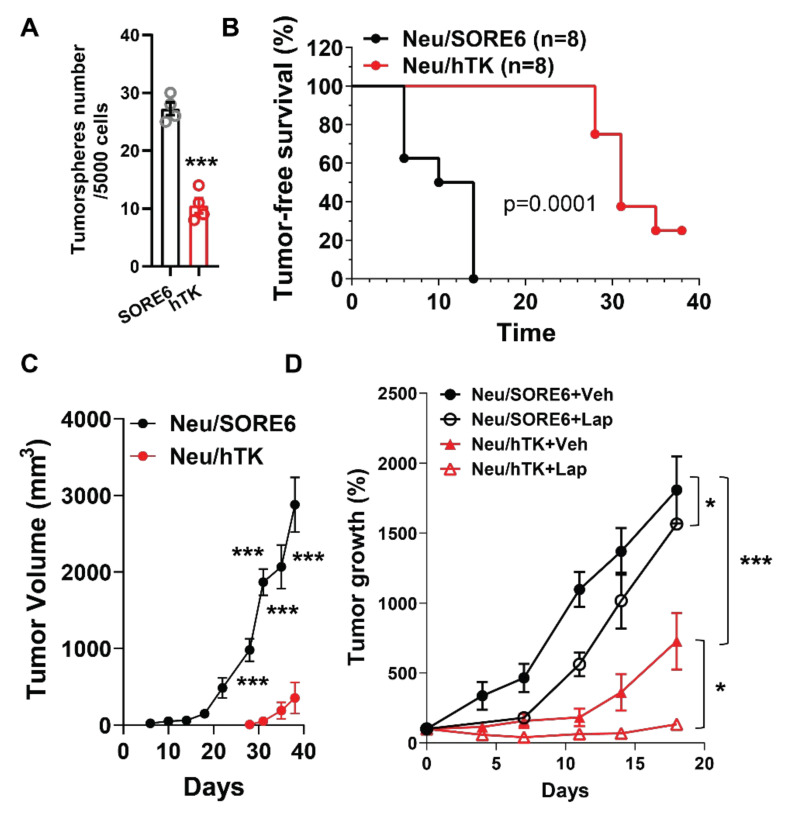
Ablation of SORE6^+^ CSCs suppresses Neu tumor initiation and progression and enhances the therapeutic efficacy of lapatinib. (**A**) GVC (10 μM) treatment suppresses tumorsphere formation of Neu cells expressing SORE6-hTK (hTK) as compared to SORE6-mCherry (SORE6). *** *p* < 0.001 vs. SORE6, *n* = 4. (**B**–**D**) Neu cells expressing SORE6-mCherry (Neu/SORE6) or SORE6-hTK (Neu/hTK) were implanted into the mammary fat pad of nude mice and were either treated with GCV (25 mg/kg, i.p., daily) for 21 days immediately following implantation (**B**,**C**) or with GCV (25 mg/kg, i.p., daily) in combination with vehicle (Veh) or lapatinib (200 mg/kg, o.g., daily; Lap) for 21 days, when tumors reached the size of 150–200 mm^3^ (**D**). Tumor growth was measured via caliper and expressed as percentage change in the tumor volume after drug treatment. (**C**) *** *p* < 0.001 vs. Neu/SORE6, *n* = 8. (**D**), * *p* < 0.05 Neu/SORE6 + Veh vs. Neu/SORE6 + Lap, or Neu/hTK + Veh vs. Neu/hTK + Lap; *** *p* < 0.001 Neu/hTK + Veh vs. Neu/SORE6 + Veh, *n* = 5–8.

## Data Availability

Not applicable.

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
