# Peer review of "Blocking Gi/o-Coupled Signaling Eradicates Cancer Stem Cells and Sensitizes Breast Tumors to HER2-Targeted Therapies to Inhibit Tumor Relapse"

_cancers, 2022, doi:10.3390/cancers14071719_

Round 1
Reviewer 1 Report
Dear author
Thank you for the submission of your article to our journal. I enjoyed your article and strongly felt findings in this paper should be an important clue to resolve the resistance of anticancer therapy for breast cancer. But I found minor mistakes in your paper as follows:
Figure 2B and 2D
Is the expression Neu/Tx correct?
Figure 4C and 4F
You should correctly express the words.
Figure 6
You should revise the word Neu/htK to Neu/hTK.
Author Response
Thank you for reviewing our manuscript and identifying several typos. We have corrected them in the revised manuscript. Specifically, we changed Neu/Tx to Neu/PTx in Figure 2B and 2D; readjusted the position of the Figure 4 and Figure 5 so that all figures and labels are correctly shown; revised the word Neu/htK in Figure 6 to Neu/hTK.
Reviewer 2 Report
In the manuscript submitted by Lyu et al, the data represent a follow through from their previous study on breast cancer published in JCI Insight Sept 2021. The data submitted here are specifically regarding the cancer stem cell compartment. The authors have used several reliable markers in multiple robust models spanning mouse models and human cancer cell lines to demonstrate their findings. The ideas are logical and coherent, data are thorough, interpretation is reasonable and experiments are well controlled and analyzed with appropriate statistics. However, there are some fundamental issues with conclusions made only using inhibitors throughout the manuscript. Additionally, there are critical formatting and typographical corrections that need to be addressed.
Major comments:
- Elaborate on Gα i/o subtype of GPCRs in the context of structural and functional distinction compared with other Gs, Gq/11, G12/13 in the introduction section for the purpose of context.
- The authors are suggested to adjust the position of Figure 4 and 5. All the data and labels are not clearly visible in the current format to make a sound judgement.
- Add “not significant” (ns) in figure 5F and G for the Akt and Src bars
- All the results presented are from pertussis toxin treatments and other inhibitors of downstream signaling pathways. This phenotype could be a result of complex effect on cellular targets. Additionally, Pertussis toxin is also shown to have both Gi/o and Gi/o independent effects on cells (Mangmool et al 2012 Toxins, https://www.ncbi.nlm.nih.gov/pmc/articles/PMC3202852/). Kindly comment and include in the discussion section.
- While majority of the data focus on the cancer stem cell entity using marker profiles and phenotype assays, more recently the generally accepted theory involves dynamic nature of cancer stem cells displaying cellular plasticity. Kindly comment. This would be important for the discussion section.
Minor comments:
- Check the spelling of
“Tumorsphere” in Fig 1D Y-axis
“Volume” in Fig 2B, 2D, 3D, 6C Y-axis
Author Response
We would like to thank you for your thorough review of the manuscript and constructive suggestions. A point-to-point response to your comments is given below.
- Elaborate on Gα i/o subtype of GPCRs in the context of structural and functional distinction compared with other Gs, Gq/11, G12/13 in the introduction section for the purpose of context.
As suggested, we have described the Gi/o-coupled GPCR signaling and function in the introduction section of the revised manuscript.
- The authors are suggested to adjust the position of Figure 4 and 5. All the data and labels are not clearly visible in the current format to make a sound judgement.
We have re-positioned the figures. Thank you for pointing this out.
- Add “not significant” (ns) in figure 5F and G for the Akt and Src bars
As suggested, not significantly different symbol was added in Figure 5F and 5G.
- All the results presented are from pertussis toxin treatments and other inhibitors of downstream signaling pathways. This phenotype could be a result of complex effect on cellular targets. Additionally, Pertussis toxin is also shown to have both Gi/o and Gi/o independent effects on cells (Mangmool et al 2012 Toxins, https://www.ncbi.nlm.nih.gov/pmc/articles/PMC3202852/). Kindly comment and include in the discussion section.
Pertussis toxin consists of the A and B protomers. While the B protomer may exert G protein-independent action by interaction with cell surface proteins, the A protomer selectively ADP-ribosylates the a subunits of Gi/o proteins. Since our transgenic mice only express the A protomer of PTx, the resulting effects on CSC activities are likely due to Gi/o-GPCR blockade. We have now included these points in the revised manuscript.
We agree that pharmacological inhibitors of PI3K/AKT and Src pathway could have out-of-target effects. However, we optimized the concentration of each inhibitor used and demonstrated their specificity in previous publication (Cancan Lyu, JCI Insight 2021). Additionally, we demonstrated the importance of these pathway in mediating Gi/o-GPCR-regulated CSC activities by genetic approaches. We have now included these points in the revised manuscript.
- While majority of the data focus on the cancer stem cell entity using marker profiles and phenotype assays, more recently the generally accepted theory involves dynamic nature of cancer stem cells displaying cellular plasticity. Kindly comment. This would be important for the discussion section.
Indeed, recent studies suggest CSCs exhibit a high level of plasticity and have the ability to switch between CSC and non-CSC states. Moreover, CSCs may consist of different sub-populations that can interconvert. Our findings that Gi/o-GPCR blockade reduces CSC populations marked by different markers suggest Gi/o-GPCR signaling supports the maintenance of multiple CSC subsets. It remains to determine whether Gi/o-GPCR signaling regulates the dynamic switch between CSC and non-CSC states or different subsets of CSCs. Nevertheless, our functional assays demonstrate the importance of Gi/o-GPCR signaling in maintaining the stemness of CSCs for tumorigenesis and therapeutic resistance in HER2+ breast cancer. We have now included these points in the revised manuscript.
Reviewer 3 Report
It would be interesting if you could present the images of the entire membrane of the blots/gels and not just cuts of them.
Author Response
Thank you for reviewing our manuscript. We have corrected several typos in the revised manuscript and provided the uncropped blots for Figure 5E in the supplemental Figure 1.